# MTS Model Application to Materials Not Starting in the Annealed Condition

**DOI:** 10.3390/ma15227874

**Published:** 2022-11-08

**Authors:** Paul Follansbee

**Affiliations:** Engineering Department, Saint Vincent College, Latrobe, PA 15650, USA; paul.follansbee@stvincent.edu

**Keywords:** deformation, constitutive behavior, strain rate dependence, temperature dependence, hardening, internal state variable, deformation kinetics

## Abstract

Application of the Mechanical Threshold Stress constitutive model becomes challenging when the material of interest is not supplied in the annealed condition with a low initial dislocation density. When the material has some existing warm or cold work, the evaluation of the internal state variables, specification of the activation energies, and analysis of strain hardening can be affected. This paper gives an example of this in molybdenum and presents options for proceeding with the model development. A hypothetical Body-Centered Cubic (BCC) alloy with known model variables is used to demonstrate the issues and solution options.

## 1. Introduction

The Mechanical Threshold Stress (MTS) constitutive model is an internal state variable formulation that relies on mechanical threshold stress $\hat{\sigma }$ values that characterize dislocation interactions with microscopic obstacle populations. These populations include grain boundaries, intentional or unintentional elemental additions, stored dislocations, or precipitates. The model has been extensively described in previous publications [1,2]. The rationale for use of an internal state variable formalism was outlined in another publication [3].

The governing equation for the temperature (*T*) and strain-rate ($\dot{\varepsilon }$) dependent stress (*σ*) in the MTS model is:


(1)
$$\frac{\sigma }{\mu }=\frac{\sigma_{a}}{\mu }+s_{p}\left(\dot{\varepsilon },T\right)\frac{{\hat{\sigma }}_{p}}{\mu_{o}}+s_{i}\left(\dot{\varepsilon },T\right)\frac{{\hat{\sigma }}_{i}}{\mu_{o}}+s_{\varepsilon }\left(\dot{\varepsilon },T\right)\frac{{\hat{\sigma }}_{\varepsilon }\left(\varepsilon ,\dot{\varepsilon },T\right)}{\mu_{o}}$$


This equation, which is written for deformation in a BCC metal, includes an athermal stress *σ_a_* (often associated with the strengthening contribution of grain boundaries), a Peierls stress (${\hat{\sigma }}_{p}$), the stress contribution of solute additions (${\hat{\sigma }}_{i}$), and the stress contribution due to interaction of dislocations with stored dislocations (${\hat{\sigma }}_{\varepsilon }$). The caret symbol specifies that these stress contributions would be the stress at 0 K, where thermal activation is ineffective. These stresses are the “mechanical threshold stresses” which comprise the internal state variables of the MTS model. The *s*-values in Equation (1) characterize the reduction of the stress required for a dislocation to overcome an obstacle due to the contribution of thermal activation. These *s*-values fall between 0 and 1. A general form for *s* follows from work of Kocks et al. [4]
(2)$$s_{j}\left(\dot{\varepsilon },T\right)={\left\{1-{\left[\frac{kT}{g_{oj}\mu b^{3}}ln\left(\frac{{\dot{\varepsilon }}_{oj}}{\dot{\varepsilon }}\right)\right]}^{1/q_{j}}\right\}}^{1/p_{j}}$$
where the subscript *j* refers to the specific obstacle population, either *p, i* or *ε* in Equation (1), *b* is the Burgers vector, *k* is Boltzmann’s constant, *g_oj_* is the normalized activation energy, and ${\dot{\varepsilon }}_{oj}$, *q_j_*, and *p_j_* are constants. Also included in these equations are the shear modulus (*µ*) and the shear modulus at 0 K (*µ*_0_).

Much of the work described in the Follansbee [2] involves application of this model to Face-Centered Cubic (FCC), BCC, and Hexagonal Close Packed (HCP) metals and alloys, austenitic stainless steels [5,6], nickel-based superalloys [7], and large-strain processed metals [8].

Application of the model and fitting of model constants requires the availability of temperature and strain-rate dependent stress–strain curves in a material with an initial low dislocation density. In this case, Equation (1) becomes:(3)$$\frac{\sigma }{\mu }=\frac{\sigma_{a}}{\mu }+s_{p}\left(\dot{\varepsilon },T\right)\frac{{\hat{\sigma }}_{p}}{\mu_{0}}+s_{i}\left(\dot{\varepsilon },T\right)\frac{{\hat{\sigma }}_{i}}{\mu_{0}}\;\;\;\;\;\;$$

Often, materials for these studies have been supplied in the recrystallized or well-annealed conditions, such that Equation (3) can be applied to the temperature and strain-rate dependent yield stress measurements. The objective of this paper is to describe potential application of the model when material is not supplied in a condition with a negligible dislocation density. For example, a material that has received a final processing step, such an elevated temperature swaging operation, will have an initial dislocation density. It will be shown that in these cases, although the MTS model cannot be applied in its rigorous form, it is possible with some assumptions to analyze hardening.

The next section will present an example of the inherent complications using published measurements in pure molybdenum. Further analysis is presented in Section 3 using the hypothetical metal-Follyalloy-introduced to illustrate the optimal application of the MTS model [2]. The case of molybdenum will be again taken up in Section 4.

## 2. Analysis of Published Measurements in Molybdenum

Molybdenum is a Body Centered Cubic (BCC) metal that has been used in commercial pure form as a model metal for deformation analyses. Briggs and Campbell [9] studied sintered molybdenum with a purity of 0.9996. The material was vacuum annealed at 1473 K for ~24 h. This yielded a material with a grain size of ~38 μm. They performed compression tests at strain rates from 1.7 × 10^−4^ s^−1^ to 100 s^−1^ and temperatures from 77 K to 600 K. They also reported measurements at a total strain of 0.08. Figure 1 shows a plot of yield stress versus temperature and strain rate according to coordinates dictated by Equation (3) with Equation (2). This plot is used to evaluate model constants in Equations (2) and (3) [1,2,10]. As described previously [10], a “two-parameter” MTS model specified by Equation (3) was applied. In this case, the athermal stress *σ_a_* was taken as 50 MPa and the threshold stress values were ${\hat{\sigma }}_{p}$ = 1541 MPa and were ${\hat{\sigma }}_{i}$ = 428 MPa. The agreement between the measurements and model predictions shown in Figure 1 is good.

Another thorough set of compression measurements in molybdenum was reported by Cheng, Nemat-Nasser and Guo [11]. These measurements were at strain rates between 0.001 s^−1^ and 3100 s^−1^ and temperatures between 300 K and 1100 K. The material was described as “commercially pure”, suggesting a purity > 0.999. However, no thermal processing condition was reported. The constitutive law developed by Cheng et al. [11] does not require that the initial dislocation density be zero, but application of the MTS model to this data set is confounded when the initial dislocation density is not zero or close to zero. This would also be the case for the application of other constitutive models. Figure 2a shows a selection of yield stress measurements in this material as a function of temperature and strain rate [11]. Included in this figure is the model line for the Briggs and Campbell measurements shown in Figure 1. Although the materials are likely of similar purity levels, the Cheng et al. measurements show higher stress levels. One can derive a best-fit model curve based on the two-parameter model; this is shown as the short-dashed line in Figure 2a. The model constants for the lines in Figure 1 and Figure 2a are listed in Table 1. The major difference in these parameters is the value of the normalized activation energy for the impurity obstacle-*g_oi_*. The value 1.5 shown is highly unusual; typically, values between 0.2 and 0.8 have been observed [2]. Thus, while one can derive model parameters, there should be little confidence in the model parameters as listed.

The likely source of the high stresses in the Cheng et al. material is that the material was supplied with some level of hot work. This might have been a final processing step, for instance, to achieve the requested bar diameter. With a material such as molybdenum, this would likely be an elevated temperature process such as swaging. The temperature, however, would be well below the recrystallization temperature. Given this level of “prework” the material would have an existing stored dislocation density, specified by ${\hat{\sigma }}_{\varepsilon }$ in Equation (1). Figure 2b shows the Cheng et al. yield stress measurements with a model fit that includes ${\hat{\sigma }}_{\varepsilon }$ = 285 MPa. Note that the other model parameters listed in Table 1 are similar to those used for the model fit shown in Figure 1.

A very plausible explanation for the higher stress levels in the Cheng et al. material is that the material was supplied with an existing stored dislocation density due to a warm working processing step. The question now becomes how to analyze the data to discern hardening behavior. This is addressed in the next section.

## 3. Analysis of Hardening in Previously Worked Material

Hardening in the MTS model arises from structure evolution due to accumulation of stored dislocations [2]. The increase of ${\hat{\sigma }}_{\varepsilon }$ is treated differentially using a modified Voce law:(4)$$\frac{d{\hat{\sigma }}_{\varepsilon }}{d\varepsilon }=\theta_{II}\left(\dot{\varepsilon }\right){\left(1-\frac{{\hat{\sigma }}_{\varepsilon }}{{\hat{\sigma }}_{\varepsilon s}\left(\dot{\varepsilon },T\right)}\right)}^{\kappa }$$
where *θ_II_* is the stage II hardening rate, e.g., of a single crystal, ${\hat{\sigma }}_{\varepsilon s}$ is the saturation value of this threshold stress and *ĸ* is a constant, usually equal to one or two. Note that the saturation threshold stress has a temperature and strain-rate dependence. This is unique from that defined for the stress in Equations (1) and (2). The kinetics are specified by a dynamic recovery model proposed by Kocks [12]:(5)$$ln\;{\hat{\sigma }}_{\varepsilon s}=ln\;\left(\;{\hat{\sigma }}_{\varepsilon so}\right)+\frac{kT}{\mu b^{3}\left(g_{\varepsilon so}\right)}ln\frac{\dot{\varepsilon }}{{\dot{\varepsilon }}_{\varepsilon so}\;}$$
where ${\hat{\sigma }}_{\varepsilon so}$ is the saturation stress at 0 K, and $g_{\varepsilon so}$ and ${\dot{\varepsilon }}_{\varepsilon so}$ are constants.

Finding these model constants requires a selection of stress–strain curves in material with an initial low dislocation density. Analysis of temperature and strain-rate dependent yield stress values gives the model constants for Equations (1) and (2)—as illustrated in Figure 1 and summarized in Table 1. Equation (1) is then solved for ${\hat{\sigma }}_{\varepsilon }$ as a function of strain. This curve can then be fit to Equation (4) to establish *θ_II_* and ${\hat{\sigma }}_{\varepsilon s}$ for the temperature and strain rate of the specified stress–strain curve. Figure 3 shows two results for measurements in zirconium at two test temperatures [2]. This is a very typical result in that *θ_II_* is not temperature dependent while ${\hat{\sigma }}_{\varepsilon s}$ has a definite temperature dependence.

With sufficient tests at various temperatures and strain rates, ${\hat{\sigma }}_{\varepsilon so}$ can be found using Equation (5). Over a very wide range of strain rates, a slight strain-rate dependence of *θ_II_* was often observed. This was described using [2]:(6)$$\theta_{II}=A_{0}+A_{1}ln\dot{\varepsilon }+A_{2}\sqrt{\dot{\varepsilon }}\;\;\;$$

When the initial dislocation density in the supplied material is suspected to be greater than zero, Equations (4)–(6) under-predict the saturation threshold stress.

The temperature dependent shear modulus, *µ*, appears in several of the above equations. The following equation proposed by Varshni [13] is used:(7)$$\mu \left(T\right)=\mu_{0}-\frac{D_{0}}{exp\left(\frac{T_{0}}{T}\right)-1}$$

At high strain rates (≥1 s^−1^) deformation is assumed to be adiabatic. The temperature rise during plastic flow is computed using:(8)$$\Delta T=\frac{\psi }{\rho c_{p}}\int \sigma d\varepsilon \;\;\;\;$$
where *ψ* is the fraction of work converted to heat (assumed to be *ψ* = 0.95), ρ is the density, and *c_p_* is the heat capacity; the temperature dependence of this physical property follows:(9)$$c_{p}=A_{C}+BT+\frac{C}{T^{2}}$$

### 3.1. Hardening in Hot Worked Follyalloy

To demonstrate the application of these equations, it was convenient to create a hypothetical alloy—Follyalloy [2]. Model constants for this alloy are listed in Table 2. The objective here is to start with well-annealed material and subject this to a hot-working operation, which strains the material 20% (*ε_hw_* = 0.20) at a temperature of 600 K (*T_hw_* = 600 K). This strain rate will be taken as 0.10 s^−1^ (${\dot{\varepsilon }}_{hw}$ = 0.10 s^−1^) which maintains isothermal conditions. From Equation (1) this operation is analyzed using
(10)$$\sigma =\sigma_{a}+s_{p}\left({\dot{\varepsilon }}_{hw},T_{hw}\right)\frac{\mu }{\mu_{0}}{\hat{\sigma }}_{p}+s_{i}\left({\dot{\varepsilon }}_{hw},T_{hw}\right)\frac{\mu }{\mu_{0}}{\hat{\sigma }}_{i}+s_{\varepsilon }\left({\dot{\varepsilon }}_{hw},T_{hw}\right)\frac{\mu }{\mu_{0}}\;\int_{0}^{\varepsilon_{hw}}\left(\frac{{\hat{\sigma }}_{\varepsilon }}{d\varepsilon }\right)d\varepsilon$$

The equation for *s_p_*, *s_i_*, and *s_ε_* was specified by Equation (2) and the equation for $d{\hat{\sigma }}_{\varepsilon }/d\varepsilon$ was specified by Equations (4)–(6). The model constants are all listed in Table 2. Equation (7) can be numerically integrated, using for instance MS Excel; strain increments of 0.001 are generally sufficient. Figure 4 shows the computed stress–strain curve for this hot-working operation. The integral on the right-hand side of Equation (10) estimates that at the end of the hot-working operation ${\hat{\sigma }}_{\varepsilon }$ = 343 MPa. That is, in the well-annealed material, this term was zero but the hot-working introduces a stored dislocation density that is characterized by this threshold stress.

This is the material that is supplied to the customer, who then performs mechanical tests as a function of temperature and strain rate. These test results can be simulated by rewriting Equation (10) as:(11)$$\sigma =\sigma_{a}+s_{p}\left({\dot{\varepsilon }}_{hw},T_{hw}\right)\frac{\mu }{\mu_{o}}{\hat{\sigma }}_{p}+s_{i}\left({\dot{\varepsilon }}_{hw},T_{hw}\right)\frac{\mu }{\mu_{0}}{\hat{\sigma }}_{i}+s_{\varepsilon }\left({\dot{\varepsilon }}_{hw},T_{hw}\right)\frac{\mu }{\mu_{0}}\;\int_{0}^{\varepsilon_{hw}}\left(\frac{{\hat{\sigma }}_{\varepsilon }}{d\varepsilon }\right)d\varepsilon +s_{\varepsilon }\left({\dot{\varepsilon }}_{2},T_{2}\right)\frac{\mu }{\mu_{0}}\;\int_{\varepsilon_{hw}}^{\varepsilon_{f}}\left(\frac{{\hat{\sigma }}_{\varepsilon }}{d\varepsilon }\right)d\varepsilon \;$$
where the threshold stress introduced by the hot-working operation (343 MPa) is added, the integral is from *ε_hw_* to the final strain in the mechanical test, and the temperature and strain rate of this mechanical test are specified as ${\dot{\varepsilon }}_{2}$ and *T*_2_, respectively.

Figure 5 shows stress–strain curves at temperatures between 100 K and 500 K and strain rates of 0.1 s^−1^ and 0.001 s^−1^. These represent the curves that the customer would measure. Figure 6 is a plot of the yield stress versus temperature (100 K to 600 K) and strain rate (0.001 s^−1^ to 1000 s^−1^). The model constants for the dashed line are listed in Table 3 in the column labeled “Material Supplied”. The “red flag” with these model constants is the high value of $g_{oi}$ (2.2). As noted in Figure 2 and Table 2 for the Cheng et al. measurements in Molybdenum, this is a very high activation energy for dislocation obstacle interactions.

Hardening can be assessed by solving for ${\hat{\sigma }}_{\varepsilon }$ in Equation (3) as illustrated for zirconium in Figure 4. Figure 7a shows the plots of ${\hat{\sigma }}_{\varepsilon }$ versus strain for three of the conditions. The dashed line model fits can hardly be discerned in these plots. The model constants for the six stress–strain curves analyzed are listed in Table 4. Figure 7b shows the variation of the saturation stress with temperature and strain rate according to Equation (5). In application of the MTS model, this plot is used to assess the model constants in Equation (5) (2). The saturation stresses decrease with increasing temperature and decreasing strain rate. The 0 K saturation threshold stress is ${\hat{\sigma }}_{\varepsilon so}$ = 490 MPa and the activation energy is $g_{\varepsilon so}$ = 0.213. The model parameters for the evolution equations are included in Table 3 in the column labeled “Material Supplied”.

### 3.2. Additional Information on the Supplied Material

Just as with the Briggs and Campbell data in annealed Molybdenum, assume that the model constants for the yield stress versus temperature and strain rate in Follyalloy were available. These would be the model constants in the top-half of Table 2 above Equations (4) and (6) rows. With analogy to the comparison between the Cheng et al. and Briggs and Campbell model parameters listed in Table 1, the researchers would note significant differences. In particular $g_{oi}$ (2.2) in Figure 6 was listed as 2.2, but this value in Table 3 is listed as 0.8. Suspecting that the material was supplied in a hot-worked condition, the researcher contacts the supplier and learns that the final processing step is a 600 K swaging operation and that the dies introduced an axial strain of 0.20.

Figure 8 shows the yield stress versus temperature and strain rate measurements for the material supplied to this researcher. The model fit now uses the Follyalloy model parameters listed in Table 2 along, but with a ${\hat{\sigma }}_{\varepsilon }$ value (343 MPa) chosen to give excellent agreement with the measured yield stresses.

With these updated model parameters, Figure 9a shows the variation of ${\hat{\sigma }}_{\varepsilon }$ with strain for three loading conditions using Equation (1). The plots start at ${\hat{\sigma }}_{\varepsilon }$ = 343 MPa, since this is the analyzed value for the hot worked material. The question remains as to how to evaluate the hardening given material that is supplied in the hot worked condition. One approach is to assume that the starting strain is that introduced by the hot working (0.20). Figure 9b shows ${\hat{\sigma }}_{\varepsilon }$ versus strain for the 298 K 0.001 s^−1^ test. Included in this plot is a model curve based on the predicted variation when the material follows the full set of model parameters specified in Table 2. The measurements fall slightly beneath the model curve at a strain level of 0.20. This is likely due to assuming that the starting strain is 0.20. In fact, for a test at this temperature and strain rate in a material starting at a strain of zero, ${\hat{\sigma }}_{\varepsilon }$ reaches 343 MPa at a strain of 0.1925, rather than at the assumed value of 0.20. Nonetheless, the error due to assuming an initial strain level of 0.20 appears to be small.

The curve shown in Figure 9b can be fit to the model evolution law, Equation (4) to estimate the fitting parameters *θ_II_* and ${\hat{\sigma }}_{\varepsilon s}$ for each of the test conditions. The resulting values are included along with the model parameters for Equations (2) and (3) in Table 4 in the column labeled “Assumed Hot Worked”. Figure 10 shows the plot of saturation stress versus temperature and strain rate according to Equation (5). There are significant differences between the evolution equation model parameters noted in the two columns. However, there are only minor differences between the model parameters for Follyalloy listed in Table 2 and those listed in Table 3 for the “Assumed Hot Worked”.

Table 3 lists two sets of model constants. Those in the column labeled “Material Supplied” do not explicitly account for evolution during the hot working operation. Those in the column labeled “Assumed Hot Worked” specify an initial value of ${\hat{\sigma }}_{\varepsilon }$ (343 MPa). Both sets of model variables will reproduce the stress–strain curve, e.g., as presented in Figure 5 and give the same values of the strain-rate sensitivity and temperature sensitivity. Figure 11 shows the measurements (solid lines) and predictions (barely visible dashed lines) for both of these sets of model parameters. The curves are offset by 200 MPa, since they essentially lay on top of each other. The “spikes” at intermediate strains are the responses in each case to a ×10 strain rate increase. Note the height of these spikes are identical in each case. Thus, both sets of model parameters accurately predict the constitutive response of this material.

This invites the question as to why bother with searching for the “correct” set of model parameters? Essential elements of the MTS model include (1) the separation of the kinetics of yield, where yield is defined as the yield stress after any loading history, from the kinetics of strain hardening, (2) consistent definitions of threshold stresses that represent microstructural strengthening mechanisms, and (3) consistent trends among model variables across all crystal structures. The first element is achieved through use of Equation (1) and (2) for the yield stress and Equation (5) for the saturation threshold stress. Application of these equations has enabled good agreement with stress–strain curves and rate dependencies. However, application of these equations does not ensure adherence to elements (2) and (3) above. It is very likely that the Material Supplied analysis has led to a ${\hat{\sigma }}_{i}$ threshold stress that combines the interaction of solute atoms with the interaction of the stored dislocation density introduced by hot working. Note that ${\hat{\sigma }}_{i}$ for the Material Supplied analysis is larger (700 MPa) than that for the Assumed Hot Worked analysis (400 MPa). A correlation between solute content and magnitude of ${\hat{\sigma }}_{i}$ has been noted in several materials. A clear dependence of ${\hat{\sigma }}_{i}$ and the carbon content was observed in iron alloys [10]. An increasing aluminum content led to an increasing magnitude of ${\hat{\sigma }}_{i}$ in titanium alloys [14]. In austenitic stainless steels, one of the mechanical threshold stresses, ${\hat{\sigma }}_{N}$, correlated well with the nitrogen concentration [6]. The interpretation of ${\hat{\sigma }}_{i}$ in the Material Supplied analysis of Follyalloy is not as clearly defined. Furthermore, note that the activation energy values listed in Table 3 are inconsistent with previous analyses. The activation energy for the Material Supplied analysis is greater (2.2) than that for the Assumed Hot Worked analysis (0.8). Recall that the large value of the activation energy for the Material Supplied analysis was noted to be unusually high for dislocation-solute obstacle interactions. The experience with a great many pure metals and alloys has typically shown this activation energy to be between 0.2 and 0.8. Finally, the model variables listed in the Material Supplied column in Table 3 do not benefit a researcher who wants to use this constitutive equation in a material that be supplied with a different starting condition. A researcher who specifically requests Follyalloy in a well-annealed starting condition could simply use the model variables listed in Table 2 with ${\hat{\sigma }}_{\varepsilon }$ = 0. Thus, there are many reasons why the model parameters listed in the Assumed Hot Worked column offer an advantage over those in the Material Supplied column.

## 4. Analysis of Hardening in Molybdenum

As discussed in Section 2, comparison between the Briggs and Campbell measurements [9] and the Cheng et al. measurements [11] in molybdenum strongly suggests the latter material was supplied in a warm worked condition. As in warm-worked Follyalloy, yield stresses measured by Cheng et al. in molybdenum are higher than those measured by Briggs and Campbell in annealed molybdenum. Furthermore, analysis of the temperature and strain-rate dependence of the yield stresses in the Cheng et al. material (Figure 2a) lead to unrealistically high estimates of the activation energy of the impurity obstacle (1.5). Assuming an initial value of ${\hat{\sigma }}_{\varepsilon }$ equal to 285 MPa, when combined with the model parameters established for the Briggs and Campbell material gives a very plausible model fit (Figure 2b).

Whereas, the hypothetical scenario proposed for the supplied Follyalloy in Section 3.2 indicated that the researcher was able to learn that the warm-working process introduced a strain of 0.20, the equivalent strain level imparted to the Cheng et al. molybdenum is not known. If it is assumed that this strain level is also equal to 0.20, one can proceed with the analysis laid out in the previous section for Follyalloy. Figure 12a shows the computed variation of ${\hat{\sigma }}_{\varepsilon }$ with strain as in Figure 9a. The strain starts at the assumed value of 0.20. The dashed lines are the model fit according to Equation (4). The curves have the predicted shape, which suggests that an assumed initial strain of 0.20 is a reasonable starting value. Figure 12b shows the plot of ${\hat{\sigma }}_{\varepsilon s}$ versus temperature and strain rate according to Equation (5) for a wide collection of test temperatures and strain rate. The model fit according to Equation (5) is drawn with a saturation threshold stress ${\hat{\sigma }}_{\varepsilon so}$ equal to 2256 MPa and the activation energy equal to 0.0852. The Stage II hardening rates all fall close to 1550 MPa.

While it was concluded, as illustrated in Figure 8, that the Cheng et al. molybdenum was supplied in a slightly worked condition with an initial ${\hat{\sigma }}_{\varepsilon }$ = 343 MPa, the strain that yielded this hardening was only estimated at 20%. The model parameters ${\hat{\sigma }}_{\varepsilon so}$ and *θ_II_* listed above depend strongly on the assumed initial strain. Figure 13a,b show the equivalent plot to Figure 12a with the initial strains assumed to be 10% (Figure 13a) and 30% (Figure 13b). The plots are all very similar. The fits to the data are perhaps better in Figure 12a and Figure 13b than in Figure 13a, but there is no way to discern the actual initial strain from these analyses. The strain induced by a warm working operation is unlikely to be less than 20%, since this may not ensure uniform deformation across the work piece, which would be undesirable. Thus, it seems likely that the strain is somewhere in the vicinity of 20% to 30%.

Table 5 lists the evolution law model parameters for all three assumed initial strains. The assumed initial strain has a large impact on these model parameters.

## 5. Conclusions

Caution is recommended when applying the MTS constitutive formalism to a material that is supplied in a worked condition. In this case, the initial dislocation density will not be negligible, which could:(1)Confound the analysis of the variation of the yield stress with temperature and strain rate. Equation (1) rather than Equation (3) is the governing kinetic equation. However, the researcher may not have sufficient information to assess the magnitude of the ${\hat{\sigma }}_{\varepsilon }$ term in this Equation (1). The strengthening contribution due to the existing stored dislocation density would be mistakenly added to the strengthening contribution, for instance, due to solute additions. Then, mechanical threshold stress values and the normalized activation energies would not represent the true strengthening contributions.(2)Introduce errors in the analysis of continued structure evolution due to dislocation accumulation. The saturation threshold stress values and the activation energy due to dynamic recovery in Equation (5) would be under estimated.

These errors can be mitigated when:(1)Prior knowledge is available of the applicable activation energies for the operative strengthening mechanisms in Equation (1), and(2)The material supplier is able to report the strains introduced during the final warm-working process.

## Figures and Tables

**Figure 1 materials-15-07874-f001:**
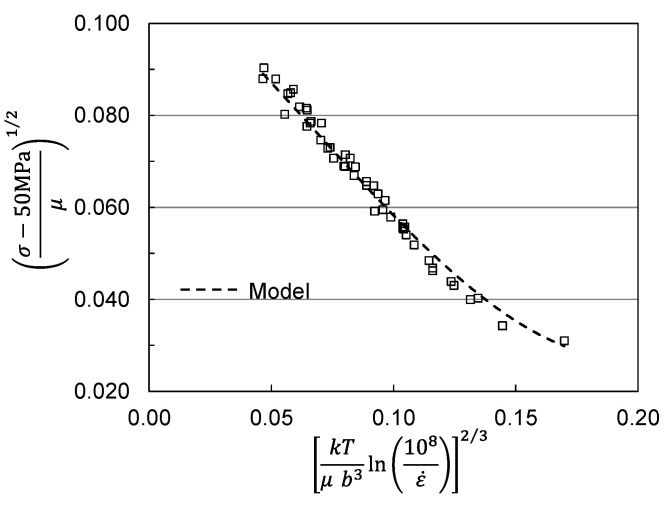
Briggs and Campbell measurements [9] in pure molybdenum analyzed according to Equations (2) and (3).

**Figure 2 materials-15-07874-f002:**
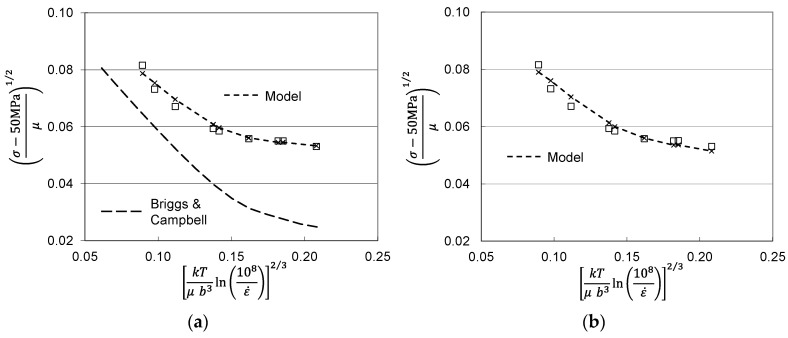
Measurements by Cheng et al. [11] analyzed according to Equations (2) and (3). (**a**) possible model fit and comparison to Briggs and Campbell fit; (**b**) model fit assuming ${\hat{\sigma }}_{\varepsilon }$ = 228 MPa.

**Figure 3 materials-15-07874-f003:**
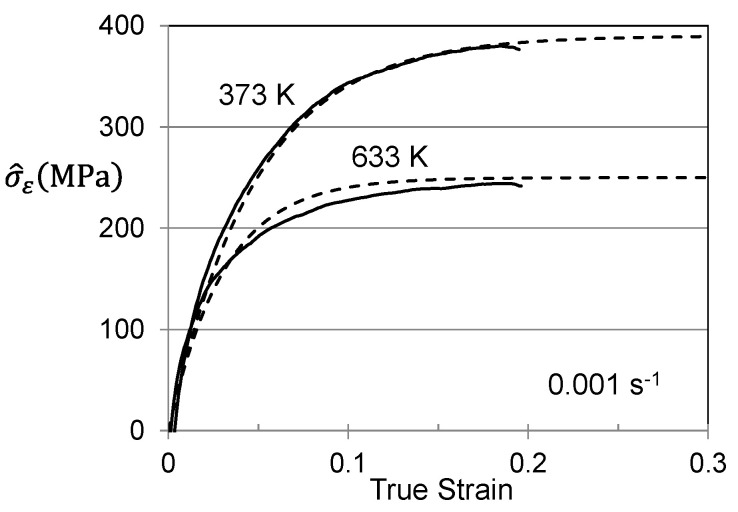
Application of Equation (4) to two measurements in zirconium.

**Figure 4 materials-15-07874-f004:**
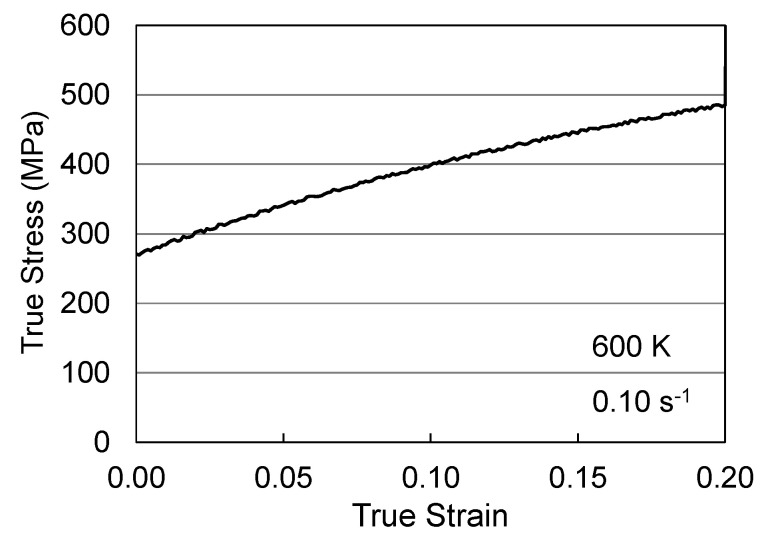
Stress–strain curve for the warm working operation in Follyalloy.

**Figure 5 materials-15-07874-f005:**
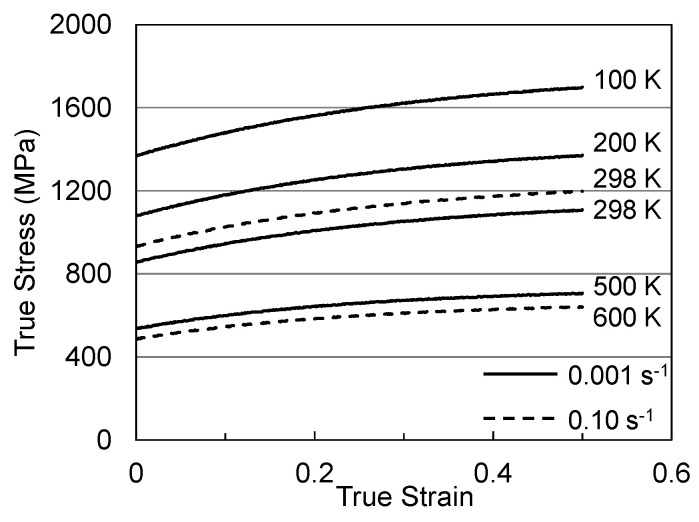
Stress–strain curves at various temperatures and strain rates in Follyalloy in the “Material Supplied” condition.

**Figure 6 materials-15-07874-f006:**
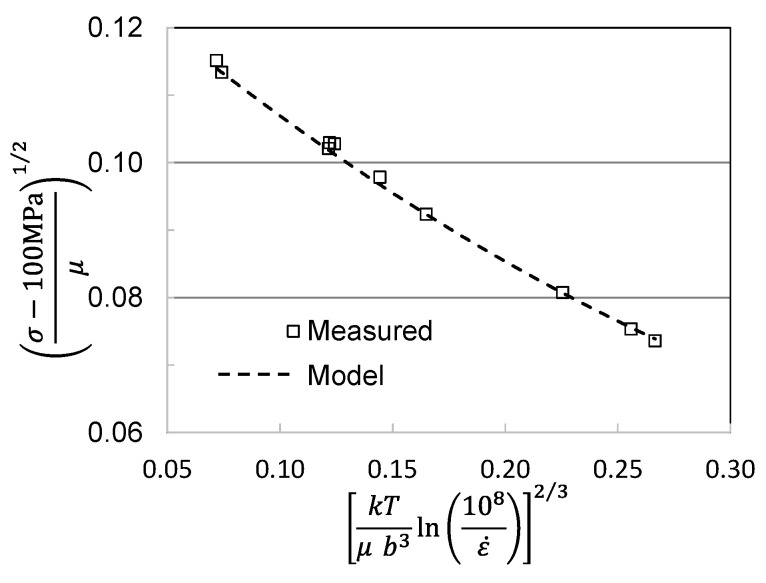
Yield stress versus temperature and strain rate in measured in Follyalloy.

**Figure 7 materials-15-07874-f007:**
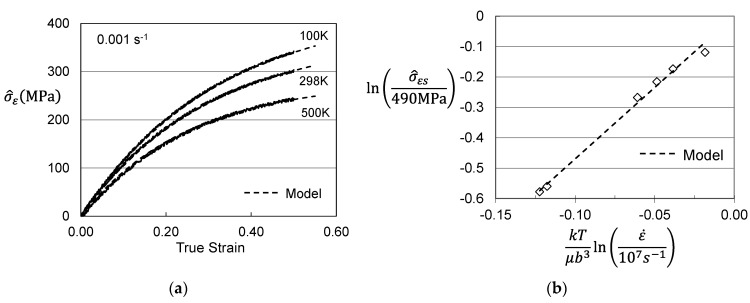
Assessed evolution for Follyalloy in the “Material Supplied” condition. (**a**) ${\hat{\sigma }}_{\varepsilon }$ versus strain according to Equation (4); (**b**) Saturation threshold stress according to Equation (5).

**Figure 8 materials-15-07874-f008:**
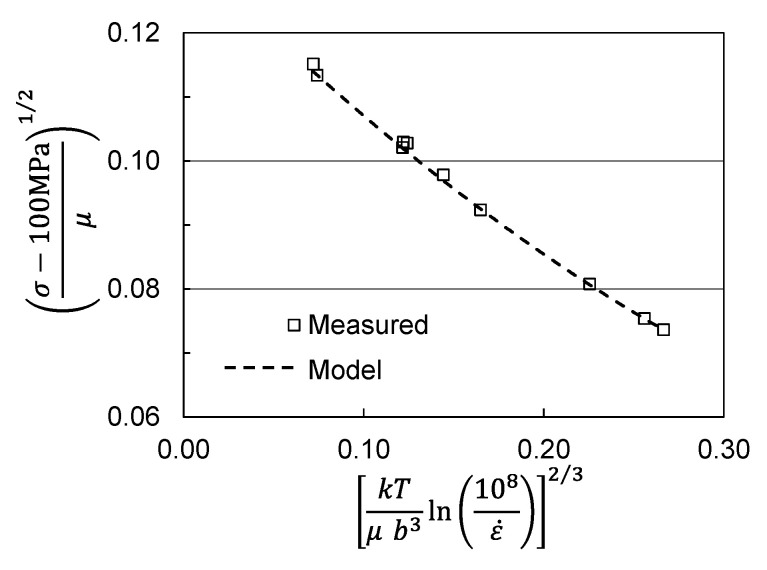
Yield stress versus temperature and strain rate in hot worked Follyalloy.

**Figure 9 materials-15-07874-f009:**
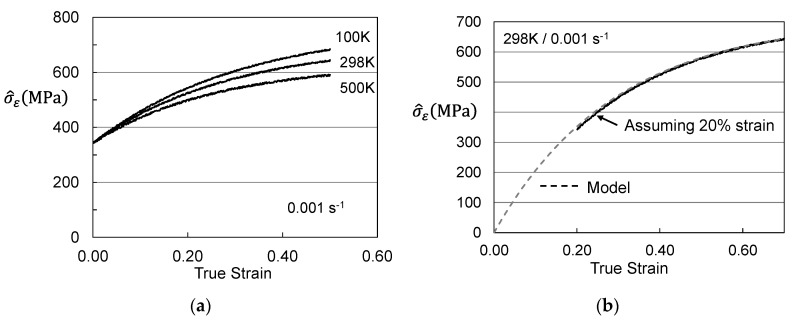
Assessed evolution for the hot worked material. (**a**) ${\hat{\sigma }}_{\varepsilon }$ versus strain according to Equation (4); (**b**) Evolution according to model versus that assuming an initial strain of 20%.

**Figure 10 materials-15-07874-f010:**
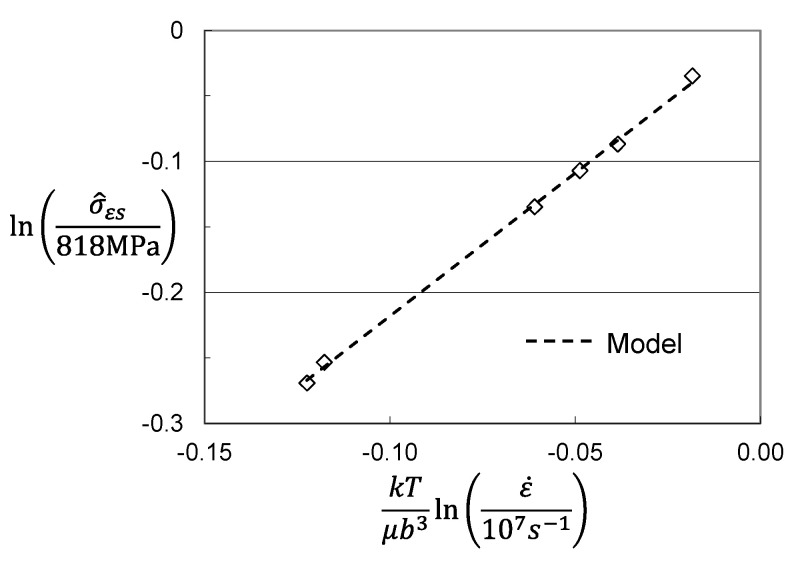
Saturation threshold stress for hot worked material with 20% initial strain.

**Figure 11 materials-15-07874-f011:**
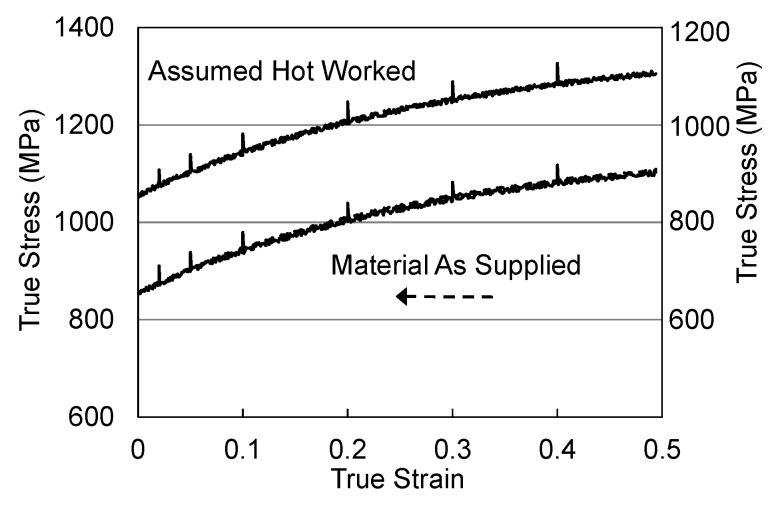
Stress–strain curves and stress response to ×10 strain rate increase.

**Figure 12 materials-15-07874-f012:**
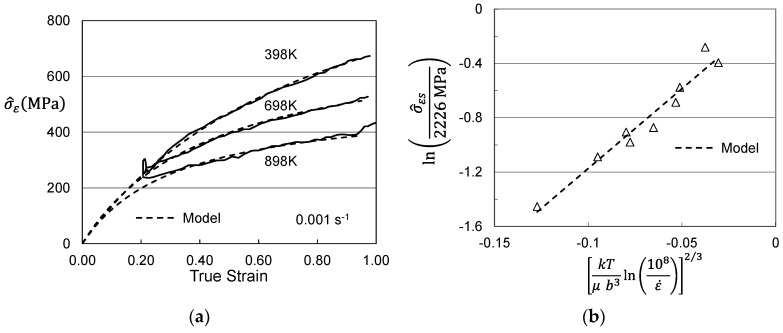
Assessed evolution for Cheng et al. molybdenum. (**a**) Variation with strain assuming an initial strain of 20%; (**b**) Saturation threshold stress according to Equation (5).

**Figure 13 materials-15-07874-f013:**
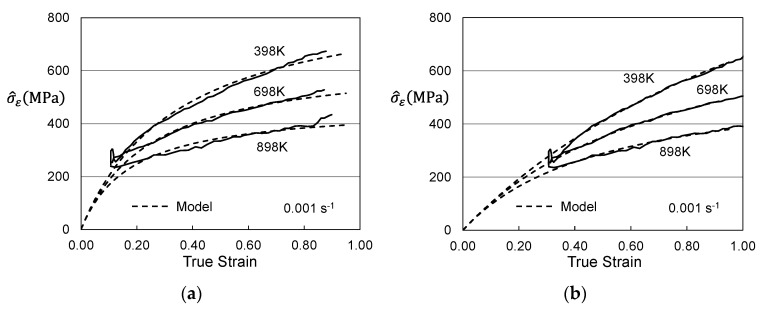
Assessed evolution for Cheng et al. molybdenum. (**a**) Variation with strain assuming an initial strain of 10%; (**b**) Variation with strain assuming an initial strain of 30%.

**Table 1 materials-15-07874-t001:** Model parameters for Figure 1 and Figure 2a,b.

Model Parameter	Briggs and Campbell [9]	Cheng et al. [11]	Cheng et al. [11]
	Figure 1	Figure 2a	Figure 2b
*σ_a_*	50 MPa	50 MPa	50 MPa
${\hat{\sigma }}_{p}$	1541 MPa	1712 MPa	1541 MPa
$g_{op}$	0.07	0.07	0.07
${\hat{\sigma }}_{i}$	428 MPa	600 MPa	513 MPa
$g_{oi}$	0.27	1.5	0.40
${\hat{\sigma }}_{\varepsilon }$	0	0	228 MPa
$g_{o\varepsilon }$			1.6
*p_p_* = *p_i_* = 0.5, *q_p_* = *q_i_* = 1.5, *p_ε_* = 0.667, *q_ε_* = 1, ${\dot{\varepsilon }}_{op}={\dot{\varepsilon }}_{oi}=10\;s^{-1}$, ${\dot{\varepsilon }}_{o\varepsilon }=10^{7}\;s^{-1}$			

**Table 2 materials-15-07874-t002:** MTS model parameters for the hypothetical Follyalloy.

	Equations (1) and (2)				
Obstacle	${\hat{\sigma }}_{j}\;(\mathrm{MPa})$	$g_{oj}$	*p_j_*	*q_j_*	${\dot{\varepsilon }}_{oi}\;(s^{-1})$
Athermal (a)	100				
Solute (*i*)	400	0.8	0.5	1.5	1 × 10^8^
Peierls (*p*)	1000	0.2	0.5	1.5	1 × 10^8^
Dislocations (*ε*)	0	1.6	0.667	1	1 × 10^7^
Physical		Equation (7)			
*b* (nm)	ρ (g/cm^3^)	*µ*_0_ (GPa)	*D*_0_ (GPa)	*T*_0_ (K)	
0.26	19.3	100	15	250	
Equations (4) and (6)					
*ĸ*	*A*_0_ (MPa)	*A*_1_ (MPa)	*A*_2_ (MPa s^−1^)		
1	2500	10	0		
Equation (5)			Equation (9)		
${\hat{\sigma }}_{\varepsilon so}$ (MPa)	$g_{\varepsilon so}$	${\dot{\varepsilon }}_{\varepsilon so}$ (s^−1^)	*A_C_* (J/g/kg)	B (J/g/K^2^)	C (JK/g)
800	0.5	1 × 10^7^	0.1345	0	0

**Table 3 materials-15-07874-t003:** Model parameters for Figures 6, 7a,b, 9b and 10.

Model Parameter	Material Supplied	Assumed Hot Worked
	Figures 6 and 7a,b	Figures 9b and 10
*σ_a_*	100 MPa	100 MPa
${\hat{\sigma }}_{p}$	1050 MPa	1000 MPa
$g_{op}$	0.2	0.2
${\hat{\sigma }}_{i}$	700 MPa	400 MPa
$g_{oi}$	2.2	0.8
${\hat{\sigma }}_{\varepsilon }$	0	343 MPa
$g_{o\varepsilon }$	1.6	1.6
${\hat{\sigma }}_{\varepsilon so}$	490 MPa	818 MPa
$g_{\varepsilon so}$	0.214	0.458
*A* _0_	1210 MPa	2410 MPa
*A* _1_	0	10
*p_p_* = *p_i_* = 0.5, *q_p_* = *q_i_* = 1.5, *p_ε_* = 0.667, *q_ε_* = 1, ${\dot{\varepsilon }}_{op}={\dot{\varepsilon }}_{oi}=10^{8}\;s^{-1}$, ${\dot{\varepsilon }}_{o\varepsilon }=10^{7}\;s^{-1}$, *ĸ* = 1, ${\dot{\varepsilon }}_{\varepsilon so}$ = 10^7^ s^−1^, *A*_2_ = 0		

**Table 4 materials-15-07874-t004:** Saturation threshold stress and *θ_II_* for the six stress–strain curves in Figure 5.

Temp (K)	Strain Rate (s^−1^)	Figure 7a $\left({\hat{\sigma }}_{\varepsilon }=0\right)$		Figure 9a $\left({\hat{\sigma }}_{\varepsilon }=343\;\mathrm{MPa}\right)$	
		Material Supplied		Assumed Hot Worked	
Entry 1		$\theta_{II}\;(\mathrm{MPa})$	${\hat{\sigma }}_{\varepsilon s}\;(\mathrm{MPa})$	$\theta_{II}\;(\mathrm{MPa})$	${\hat{\sigma }}_{\varepsilon s}\;(\mathrm{MPa})$
298	0.001	1230	375	2350	715
200		1270	412	2320	750
100		1320	435	2290	790
500		1120	280	2380	635
298	0.10	1270	395	2340	735
600		1140	275	2430	625
		*ĸ* = 1			

**Table 5 materials-15-07874-t005:** Evolution model parameters for the three assumed initial strains.

Model Parameter	Assumed Initial Strain		
	0.20	0.10	0.30
	Figure 12a	Figure 13a	Figure 13b
${\hat{\sigma }}_{\varepsilon so}$	2226 MPa	1432 MPa	3285 MPa
$g_{\varepsilon so}$	0.0852	0.116	0.0732
*A* _0_	1550 MPa	2600 MPa	1100 MPa
*A* _1_	0	0	0

## Data Availability

All of the data presented is available in the citations listed.

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
