# Peer review of "MTS Model Application to Materials Not Starting in the Annealed Condition"

_materials, 2022, doi:10.3390/ma15227874_

Round 1
Reviewer 1 Report
The author presented the potential use of the Mechanical Threshold Stress (MTS) constitutive model of material under not annealed condition with a low initial dislocation density.
The paper seems interesting, despite its theoretical character only. I have no main doubt about its content, although, on the other hand, theoretical considerations not supported by experiments may not be fully reliable.
I found some errors in this manuscript and it should be improved.
Weak
1. The biggest weakness of this paper seems to be the slightly underdeveloped graphic part, especially the charts. Minor errors should be corrected, and charts details harmonized.
Noticed errors
1. While reading the paper, I had doubts about which charts were the author's and which were from other sources, since they lacked literature references. It would be appropriate to specify this according to the rules of art.
2. Chapter 5. There is a lack of conclusions for further research, the addition of which would certainly increase the value of the work.
3. References. The chapter has only 14 items which, even for an article of a theoretical nature, does not knock it down on the timber. Half of it is self-citations, which is also not common in scientific articles. My doubts are also raised by the not latest items on the list. Only two sources are very new, the rest are from 8 years ago.
Small errors
These errors do not diminish the value of this interesting work, but need to be improved
1. Figure 1, 2, 6, 7, 8, 9, 10, 12, 13. The MPa is italic on OY axis. Is it the variable? If not, it must be standard font.
2. Line 72-74. Paragraph must be aligned.
Reviewer 2 Report
Reviewing Report
This work describes the potential application of the MTS model when material is not supplied in a condition with a negligible dislocation density. It will be a good reference material for researchers in the field of engineering. I recommend the publication after affecting these minor corrections in the manuscript.
1. Ln 53 -54, pp. 2 “This may be the case is the final processing of rod material, for instance, is an elevated temperature swaging operation” Kindly rephrase this sentence for clarity”.
2. Ln 62 - 63 “Molybdenum is a Body Centered Cubic (BCC) metal that has been used in commercial pure for as a model metal for deformation analyses”. Do you mean commercial pure “for” or “form”
3. Ln 96 – 97, pp. 3 “The value 1.5 shown in highly unusual; typically, values between 0.2 and 0.8 have been observed (2). Do you mean “in highly unusual” of “is highly unusual”
4. Ln 103, pp. 4 “this would like be an elevated temperature process such as swaging” should read “this would be like an elevated temperature process such as swaging”
5. Ln 141, pp. 5 “This was describe using”. Should read “This was described using”
6. Ln 154, pp. 5 “Where Ψ is the fraction of work converted to hear”. Check if you mean “hear” or “heat”
7. Ln 258, pp. 9 “...versus temmperature and strain rate according to Equation (5)”. Correct the spelling of “temperature”
8. Ln 259, p.9 “... differences between the evolution equation model equations noted in the two columns”. Check the sentence and make necessary correction.
9. Ln 299 – 300, pp. 10 “The activation energy for the Material Supplied analysis is greater (2.2) that for the Assumed Hot Worked analysis (0.8)”. Check if “than” is not missing in that sentence.
10. Apart from works by the author ( references 2 and 8) which were done in 2022, all other citations are too obsolete. Does it mean that no recent work has been done in this work by other researchers?
Reviewer 3 Report
In my opinion, the subject of the paper is interesting. Besides, the article is well organized. However, the paper needs some revisions. Finally, I recommend the paper for publication after the following revisions are performed:
(1) The paper includes some minor typos, punctuation, and grammatical errors. The whole paper should be checked.
(2) The novelty of this work should be more discussed. Authors may explain deficiencies or shortcomings of other studies to make a bridge to introducing the novelty of their work.
(3) The advantage of the proposed experimental methodology should be more discussed.
(4) Authors should add some physical explanation to improve the quality of the paper. The conclusion, introduction and abstract sections should be extended via the main finding and advantages of the methodology.
